# Laryngopharyngeal Reflux Diagnosis in Obstructive Sleep Apnea Patients Using the Pepsin Salivary Test

**DOI:** 10.3390/ijerph16112056

**Published:** 2019-06-10

**Authors:** Giannicola Iannella, Claudio Vicini, Antonella Polimeni, Antonio Greco, Riccardo Gobbi, Filippo Montevecchi, Andrea De Vito, Giuseppe Meccariello, Giovanni Cammaroto, Giovanni D’Agostino, Annalisa Pace, Raffaella Cascella, Marco Brunori, Cristina Anna Maria Lo Iacono, Stefano Pelucchi, Giuseppe Magliulo

**Affiliations:** 1Department of ‘Organi di Senso’, University “Sapienza”, Viale dell’Università 33, 00161 Rome, Italy; antonio.greco@uniroma1.it (A.G.); annalisapace90@gmail.com (A.P.); raffaellacascella@virgilio.it (R.C.); giuseppe.magliulo@uniroma1.it (G.M.); 2Department of Head-Neck Surgery, Otolaryngology, Head-Neck and Oral Surgery Unit, Morgagni Pierantoni Hospital, Via Carlo Forlanini, 34, 47121 Forlì, Italy; claudio@claudiovicini.com (C.V.); dr.riccardogobbi@gmail.com (R.G.); filippomontevecchi72@gmail.com (F.M.); dr.andrea.devito@gmail.com (A.D.V.); drmeccariello@gmail.com (G.M.); giovanni.cammaroto@hotmail.com (G.C.); giovanni.dagostino77@gmail.com (G.D.); 3Department ENT and Audiology, University of Ferrara, Via Aldo Moro, 8, 44124 Ferrara, Italy; stefano.pelucchi@unife.it; 4Department of Oral and Maxillo Facial Sciences, University “Sapienza”, Via Caserta, 6, 00161 Rome, Italy; antonella.polimeni@uniroma1.it; 5Department of Cardiovascular, Respiratory, Nephrologic, Anaesthesiologic and Geriatric Sciences, Sapienza University, Viale dell’Università 33, 00161 Rome, Italy; marco.brunori@uniroma1.it (M.B.); cristina.loiacono@uniroma1.it (C.A.M.L.I.)

**Keywords:** laryngopharyngeal reflux, obstructive sleep apnea, Obstructive Sleep Apnea Syndrome, PEP-test

## Abstract

Background: To investigate the presence of laryngopharyngeal reflux in patients with obstructive sleep apnea (OSA) employing the salivary pepsin concentration method. To compare the results of pepsin concentration with the severity of the pathology. Methods: Seventy-five OSA patients (44 males, 31 females) were enrolled in the study. For each patient, the AHI (apnea–hypopnea index) and the BMI (body mass index) were initially evaluated. All the patients enrolled were assessed using the reflux symptom index (RSI) and the reflux finding score (RFS) in order to perform a clinical diagnosis of laryngopharyngeal reflux. In all patients a salivary sample was taken to estimate the presence of pepsin and its concentration. Results: The incidence of LPR (laryngopharyngeal reflux) in OSA patients, evaluated using the salivary pepsin concentration test (PEP-test), was found to be 32% of cases. Linear regression testing did not show any correlation between AHI and pepsin concentration in salivary samples (*p* = 0.1). Conclusion: A high number of patients with OSA seem to show positivity for salivary pepsin, correlated to an LPR. There does not appear to be a correlation between the severity of apnea and the grade of salivary pepsin reflux. On the other hand, direct correlation between BMI and the value of pepsin in salivary specimens was observed.

## 1. Introduction

Obstructive sleep apnea (OSA) is a common health problem affecting about 5% of the adult population [1,2]. It is a sleep disorder characterized by multiple episodes of airflow obstruction due to the collapse of different upper airway structures [1,2,3,4]. Abnormalities in upper airway anatomy (tonsils, soft palate, base of the tongue, and/or hypopharynx) as well as neuromuscular control are the primary factors that contribute to upper airway collapsibility in OSA patients [2,3,4,5,6,7,8]. The severity of the OSA syndrome is mainly measured using the AHI (apnea–hypopnea index) which is based on the number of apnea/hypopnea events per hour of sleep.

Over recent years, various studies have been published dealing with the correlation between laryngopharyngeal reflux (LPR) and OSA [9,10,11,12,13,14,15,16,17]. Although a high incidence of LPR in this class of patients has been reported [9,10,11,12,13,14,15,16,17], the possibility of a correlation between the severity of the pathology (number of hypopnea/apnea episodes) and LPR has not yet been clearly demonstrated [10,11,12,13,14,15,17].

Magliulo et al. [18] confirmed this data in a recent meta-analysis of the international literature: They found a high incidence of LPR (45.2%) in OSA patients but did not observe any correlation between the grade of AHI (apnea–hypopnea index) and the presence of LPR. As emerged from this meta-analysis, in most of the published studies the LPR diagnosis was based on two validated questionnaires developed by Belafsky et al. [19,20], namely the reflux symptom index (RSI) and the reflux finding score (RFS). These are indirect tests based on clinical and endoscopically-observed characteristics. The gold standard for demonstrating reflux events are multi-channel intraluminal impedance (MCII) and pH monitoring studies [21,22,23]. Unfortunately, these tests are expensive, not easily applicable, and are often either not accepted or tolerated by patients [23,24,25]. Therefore, in clinical practice, a diagnosis based on endoscopic findings and clinical questionnaires is usually preferred [9,11,20,23,24,25].

Recently, the salivary pepsin concentration test (PEP-test) has been introduced into clinical practice and a number of authors have supported its use for diagnosing LPR disease [26,27,28,29,30]. Barona-Lleo et al. [31], as well as other authors [27,28,29], have confirmed that the PEP-test is a simple, inexpensive, and easily reproducible test that should be considered as an alternative diagnostic tool for LPR diagnosis. Moreover, it avoids the need for empirical treatment or other invasive tests.

The main aim of the present study is to investigate the presence of LPR in patients with OSA via salivary pepsin concentration. The secondary objective is to compare the results of pepsin concentration to the severity of the pathology (number of hypopnea/apnea episodes).

## 2. Materials and Methods

### 2.1. Subjects of the Study

This prospective bi-center study was performed at the Otolaryngology, Head and Neck and Oral Surgery Department of the Morgagni Pierantoni Hospital in Forlì, Italy and at the ‘Organi di Senso’ Department of ‘Sapienza’ University in Rome, Italy, between January and September 2018.

Subjects eligible for the study were initially selected from patients referred to these departments in whom OSAS was suspected. All these patients underwent polysomnography (PSG) after the initial evaluation in order to obtain a diagnosis of this pathology.

The American Academy of Sleep Medicine (AASM), defines OSAS as the presence of multiple episodes of airflow obstruction (at the PSG examination) during sleep due to the collapse of different upper airway structures.

In accordance with the AASM, the diagnosis and classification of OSAS was based on the apnea–hypopnea index (AHI) [32,33]. Patients were classified as either normal (AHI was <5/h), mild OSAS (AHI ≥ 5 and AHI < 15), moderate OSAS (AHI ≥ 15 and AHI < 30) or severe OSAS (AHI ≥ 30) [32,33]. The simple snorers according to the results of PSG (AHI < 5/h) were excluded from the study [32,33].

The exclusion criteria applied were: age < 18 years old, presence of oral diseases and use of systemic steroids, pump inhibitors or treatment with other drugs for LPR (laryngopharyngeal reflux disease) at the time of the study. None of the patients enrolled were undergoing treatment for OSAS with a continuous Positive Airway Pressure (c-PAP) device or other medical devices (e.g., oral appliance). Finally, patients who had undergone previous sleep apnea surgery were excluded from the study.

All patients gave their written consent for all the tests and for their enrolment in the study.

This research was performed in accordance with the principle of the Declaration of Helsinki and approved by the local Ethics Committee of the Morgagni Pierantoni Hospital of Forlì and “Sapienza” University of Rome (Ethical approval number: RIF. CE 4841 19-01-2018).

### 2.2. Clinical and Endoscopic Investigation of LPR

Clinical data, including height and weight, in order to calculate body mass index (BMI), were initially collected for each patient enrolled in the study.

All patients initially considered for inclusion in the study underwent reflux symptom index (RSI) evaluation and reflux finding score (RFS) evaluation [19,20]. 

The RSI is a self-conducted questionnaire, developed by Belafsky [19,20], based on nine questions, with a maximum of 5 points for each answer, giving a total maximum score of 45 points. As indicated by Belafsky and validated by other authors, the score was considered pathological when it was ≥13 [19,20,34,35].

The RFS evaluates the presence of 8 laryngoscopic findings, with a scale going from 0 (normal) to 26 (strongly pathological). RFS ≥ 7 is considered pathological and indicative of LPR [19,20]. The same author (G.I.) calculated the RFS score in all enrolled patients in order to obtain homogeneous results for RFS evaluations. Besides, he did not know the results of the RSI, thus limiting possible confounding factors (blinded data). The endoscopic evaluation was performed using a flexible endoscope connected to a camera and a high-definition monitor (Full HD).

Patients with RSI and RFS positivity were considered clinically positive for LPR. Due to the many diagnostic weaknesses of these clinical tests, only patients with pathological scores in both the RSI and RFS evaluations were classified as clinically positive for LPR in this study [19,20]. Patients with pathological scores in only one of these two investigations were excluded from the study in order to reduce a false positive in LPR diagnosis obtained with a single positive test.

### 2.3. Salivary Specimen Collection and Salivary Pepsin Concentration Estimation (PEP-Test)

In all patients enrolled in the study a salivary sample was taken in order to estimate the salivary presence of pepsin and its concentration. Patients with positive salivary pepsin were considered positive for LPR.

Salivary specimen collection and PEP-test measurement were performed in each patient in a range between 5 days and 21 days after PSG examination, the mean time interval that elapsed between the PSG and PEP-test investigation was 8.9 days.

The salivary samples were collected in the early hours of the morning (7.00–8.00 a.m.) with the patient in an upright position. A micropipette, with a 0.3 cm diameter silicone tube, 2 cm long and cut obliquely at 45°, siliconized to a small tank (diameter 0.5 cm, 2 cm long), provided with a suction tube, was employed to directly pick up the saliva from the oral cavity.

The salivary samples were analyzed using a PEP-test^TM^ kit (BIOHIT HealthCare) immediately after the salivary collection in order to avoid damaging the specimen.

The PEP-test is a qualitative and quantitative in vitro immunological test for dosing the pepsin concentration in body fluids [26,27,28,29].

It is a very accurate test with a validated specificity of 87% and a sensitivity of 88% with a pepsin detection limit of 16 ng/mL [26,27,28,29,30,31].

The test required 100 µL of salivary sample with the addition of 100 µL of 0.01 M citric acid.

Each sample was centrifuged at 400 rpm for 5 min at normal room temperature. Subsequently, 80 µL of supernatant was collected and added to 240 µL of migration buffer. The mixture was then vortexed for 10 s: 80 µL of this mixture was pipetted into the well of the Pep-test^TM^ Lateral Flow Device (LFD) and the results were ready after 15 min.

The test is based on an immunohistochemical reaction (antigen-antibody reaction) utilizing a monoclonal anti-pepsin antibody (the T band reveals the pepsin presence). In addition, the system involves an inner reaction control useful for estimating the system’s integrity (C band). The test is valid when it obtains a reaction related to the internal control (IC, C band). The existence of the T band indicates that pepsin is present in the tested sample and, furthermore, the intensity of the T band is directly proportional to the pepsin quantity (Figure 1).

The pepsin concentration level was accurately measured using the PEP-test Cube. This is a small electronic lateral flow device that displays the result of pepsin concentration analysis in different fluids directly in ng/mL in just a few seconds (Figure 2).

The Cube is suitable for use in the quantitative and qualitative evaluation of pepsin measurement results and it displays the result of pepsin concentration directly in ng/mL in just three seconds.

Weighing only 40 g and with an edge length of 41 mm, the PEP-test Cube can be used as a mobile handheld device or as a desktop measuring device remotely controlled via USB cable.

The reader is able to detect a minimum amount of pepsin concentration equal to 16 ng/mL. The device is able to conduct colorimetric tests based on reflectance measurements that capture the optical density. The test can provide three possible results: negative (only the IC is present), positive (the T and C bands are present), null (absence of IC signal).

### 2.4. Statistical Analysis

Statistical analysis was performed using Statview statistical software version 8.0.

Mean age, mean AHI, and mean BMI of LPR positive (+) and LPR negative (−) patients were compared using the Student’s *t*-test. A value of *p* < 0.05 was considered statistically significant.

The Chi-square test was used to evaluate differences in OSAS severity classes and LPR positivity. A value of *p* < 0.05 was considered indicative of statistical significance.

Linear regression testing was carried out to evaluate a possible correlation between AHI and pepsin concentration in addition to BMI and pepsin concentration in salivary samples. Linear regression was also employed to evaluate a possible correlation between AHI and RFS and RSI results. In this case a value of *p* < 0.05 was considered statistically significant.

## 3. Results

Seventy-five OSA patients (44 males, 31 females; 19–75 years of age, average 50.9) were enrolled in the study after application of the inclusion/exclusion criteria.

### 3.1. LPR Prevalence in OSA Patients

Patients positive in RSI, RFS, and PEP-test evaluations are shown in the histogram (Figure 3).

A clinical diagnosis of LPR (both RSI and RFS positivity) emerged in 36 (48%) of studied OSA patients.

The average RSI value in these patients was 19.1 (13–29) and the median value was 18.5.

Regarding the RFS score, the average value was 10.2 (8–15) and the median value was 10.

The pepsin evaluation in the salivary sample showed positivity in 24 of the 75 OSA patients enrolled in the study. Hence, the incidence of LPR in OSA patients evaluated by means of the PEP-test was calculated to be 32% of cases.

All patients with a positive PEP-test also showed positivity in RSI and RFS scores, whereas 12 (16%) patients showed positivity in the clinical investigation but a negative PEP-test (Table 1). Therefore, the clinical diagnosis of LPR (performed using RSI and RFS scores) was confirmed by means of the PEP-test in 66.6% of these patients.

None of the investigated patients showed PEP-test positivity and RSI and RFS negativity.

The average level of pepsin concentration in the 24 PEP-test-positive patients was 34.1 µg/mL (16.9–54.7 µg/mL).

### 3.2. LPR and Age

No statistically significant differences emerged between the LPR+ and LPR− (PEP-test diagnosis of LRP) patients in relation to age (Table 1).

### 3.3. LPR and AHI

The mean AHI value of LPR+ patients (PEP-test diagnosis of LRP) and the mean AHI value of LPR– patients did not show any statistical difference (*p* = 0.2) (Table 1).

Linear regression failed to show a correlation between AHI and pepsin concentration in salivary samples (*p* = 0.19, R^2^ = 0.07) (Figure 4).

Similarly, linear regression did not show any correlation between values of AHI and RSI (*p* = 0.4, R^2^ = 0.02), nor AHI and RFS (*p* = 0.6, R^2^ = 0.01).

Similarly, no differences emerged in a comparison of the AHI classes of OSA severity and LPR+ and LPR− patients (*p* > 0.05 for each class of OSA severity, chi square test).

### 3.4. LPR and BMI

Mean BMI was higher in patients with salivary pepsin positivity than in PEP-test negative patients, with a statistical difference (*p* = 0.02; Table 1) (Figure 5). Moreover, a direct correlation between BMI and the value of pepsin in salivary specimens was submitted to regression analysis (*p* = 0.05 R^2^ = 0.2; Figure 6).

## 4. Discussion

Patients with OSA suffer from LPR far more frequently than those of the general population [12,13,14,15]. The causative relationship between obstructive events and LPR disease could depend on a vicious circle [36]. The vicious cycle is initially triggered by respiratory efforts. As airflow obstruction develops in OSA, the progressive increase in respiratory effort produces greater negative intrathoracic pressure. As this negative intrathoracic pressure exceeds the ability of the containment function of the lower esophageal sphincter, reflux of gastric contents (acid, pepsin, etc.) into the esophagus, larynx, and pharynx occurs. Contrarily, LPR creates inflammation and sensory deficits in the laryngeal and pharyngeal tissues that contribute to progression of OSA via both inflammatory and neuromuscular pathways [18,36].

Therefore, it is clear that formulating a diagnosis of LPR is fundamental in these patients.

Today, in the majority of cases, LPR is suspected on the basis of clinical observations and diagnosis is based on the presence of symptoms suggestive of this disease. Specific clinical and endoscopic validated questionnaires (RSI and RFS) have been proposed and validated in order to obtain an LPR diagnosis and to avoid invasive tests such as pH-metry [23,24,25,26,27,28,29,30,31,32,33,34,35,36,37]. Although these questionnaires are simple to administer to patients, as diagnostic tools they have many weaknesses. LPR symptoms evaluated using the RSI may be non-specific, since they can be found among subjects without reflux [23]. Chen et al. [37] observed a significant rate of throat clearing, excess throat mucus or postnasal drip, and globus sensation among healthy subjects. Secondly, the RSI assesses the severity of LPR complaints by means of a visual analogue scale but does not take into consideration the frequency of symptoms. Regarding RFS findings, 80% of healthy subjects could have ≥1 signs of laryngopharyngeal irritation, including laryngeal erythema, posterior commissure hypertrophy, or diffuse laryngeal edema.

In accordance with these findings it is possible to understand how this indirect test might provide misleading information and erroneous LPR diagnosis, especially in cases of concomitant or overlapping pathologies.

Recently, a new diagnostic tool for LPR diagnosis has been developed and marketed. It is a rapid immunohistochemical test (PEP-test), capable of detecting the pepsin concentration in salivary specimens [23,24,25]. Different studies have suggested that the presence of pepsin in the pharynx is a direct expression of reflux episodes [26,27,28,29,30,31]. Consequently, the PEP-test has been considered to be a reliable diagnostic marker of LPR with a strong predictive value for the diagnosis of LPR [23].

In the light of this, Wang et al. [26] performed a prospective study, enrolling patients who had presented laryngopharyngeal reflux (RSI > 13) for more than three months and treating them with 40 mg of esomeprazole for eight weeks. They demonstrated a correlation between strongly positive results at the PEP-test (*p* < 0.05) associated with a good protonic pump inhibitor (PPI) response with a predictive value of 79.2%.

A number of recent studies have investigated the prevalence of reflux in OSA patients but in almost all cases this was done using clinical questionnaires alone [9,10,11,12,13,14,15,16,17]. These aspects emerged in a recent meta-analysis of the international literature, which analyzed the incidence of LPR in OSA patients [18]. All studies included in the meta-analysis published by Magliulo et al. [18] evaluated the incidence of LPR in OSA patients using clinical questionnaires (RSI and RFS).

In the present study, the LPR evaluation in OSA patients was conducted using PEP-test evaluation. The prevalence of LPR in OSA patients estimated using the PEP-test was 32%. Differently, clinical diagnosis using the RSI and RFS was performed in 48% of patients.

All patients with positive PEP-test results also showed positivity in RSI and RFS scores, whereas 12 (16%) patients showed positivity in the clinical investigation but a negative PEP-test. The diagnosis of LPR was confirmed using the PEP-test in 66.6% of patients with positive RSI and RFS evaluations.

Linear regression did not show any correlation between AHI values and pepsin concentration in salivary samples (*p* = 0.1). Similarly, linear regression failed to show any correlation between values of AHI and RSI (*p* = 0.4) or of AHI and RFS (*p* = 0.6).

These results seem to confirm the results previously reported in the literature [12,13,16,18], regarding statistical differences in the AHI values between LPR+ and LPR− patients.

BMI was higher in patients with PEP-test positivity and LPR diagnosis. Linear regression between BMI and salivary pepsin concentration showed a positive correlation of these two parameters.

As pointed out by several authors, in the general population a high BMI could favor laryngopharyngeal reflux events. Our data seemed to support this possibility in OSA patients too, in accordance with the results of the meta-analysis study carried out by Magliulo et al. [18].

The main limitation of this study is the lack of pH-metry evaluation in OSA patients.

Twenty-four-hour double probe pH-monitoring is considered the gold standard for diagnosis of GERD (gastroesophageal reflux disease); however, pH-metry has a low sensitivity for LPR (50%–80%) [25,27]. Besides, PH-metry is invasive, expensive, and could be misrepresented by PPI and diet therapy.

Another limitation is the restricted number of enrolled patients, although other studies with a larger series of patients are under way. In additions, further clinical studies are currently going on to confirm these clinical results, using pH-metry too.

## 5. Conclusions

A high number of patients with OSA seem to show positivity for salivary pepsin, correlated to LPR. The severity of apnea does not seem to correlate with the entity of salivary pepsin reflux. However, further clinical studies including a control group must be conducted to confirm these clinical results.

## Figures and Tables

**Figure 1 ijerph-16-02056-f001:**
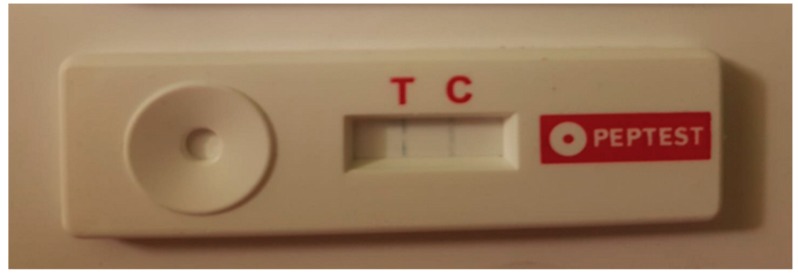
The salivary pepsin concentration test (PEP-test): The existence of the T band indicates that pepsin is present in the tested sample and, furthermore, the intensity of the T band is directly proportional to the pepsin quantity.

**Figure 2 ijerph-16-02056-f002:**
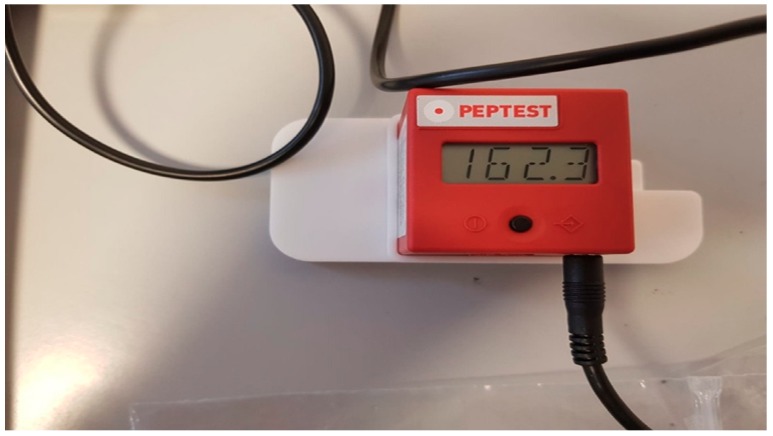
The PEP-test Cube: A small electronic device that displays the result of pepsin concentration analysis in different fluids directly in ng/mL.

**Figure 3 ijerph-16-02056-f003:**
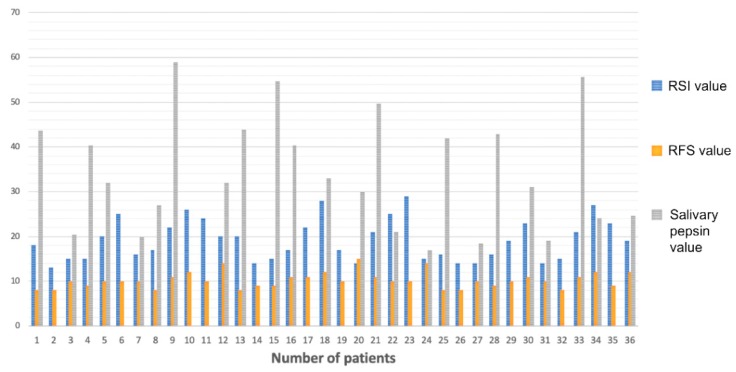
Histogram: Patients positive in reflux symptom index (RSI) (blue column), reflux finding score RFS (yellow column), and PEP-test (gray column) evaluations. Patients with only RSI and RFS columns were those that showed positivity to clinical investigation but negativity for salivary pepsin presence (negative PEP-test).

**Figure 4 ijerph-16-02056-f004:**
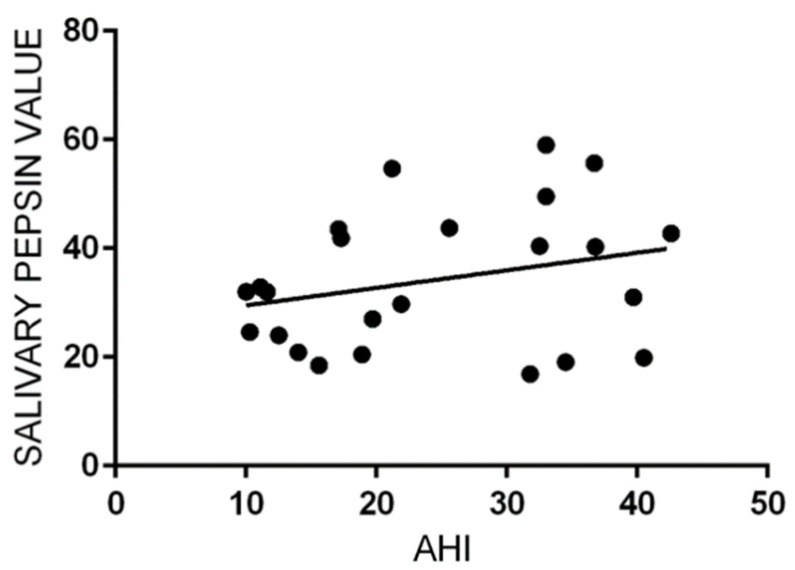
Liner regression scatterplot between apnea–hypopnea index (AHI) and salivary pepsin concentration. No statistical correlation resulted (*p* = 0.19, R^2^ = 0.07).

**Figure 5 ijerph-16-02056-f005:**
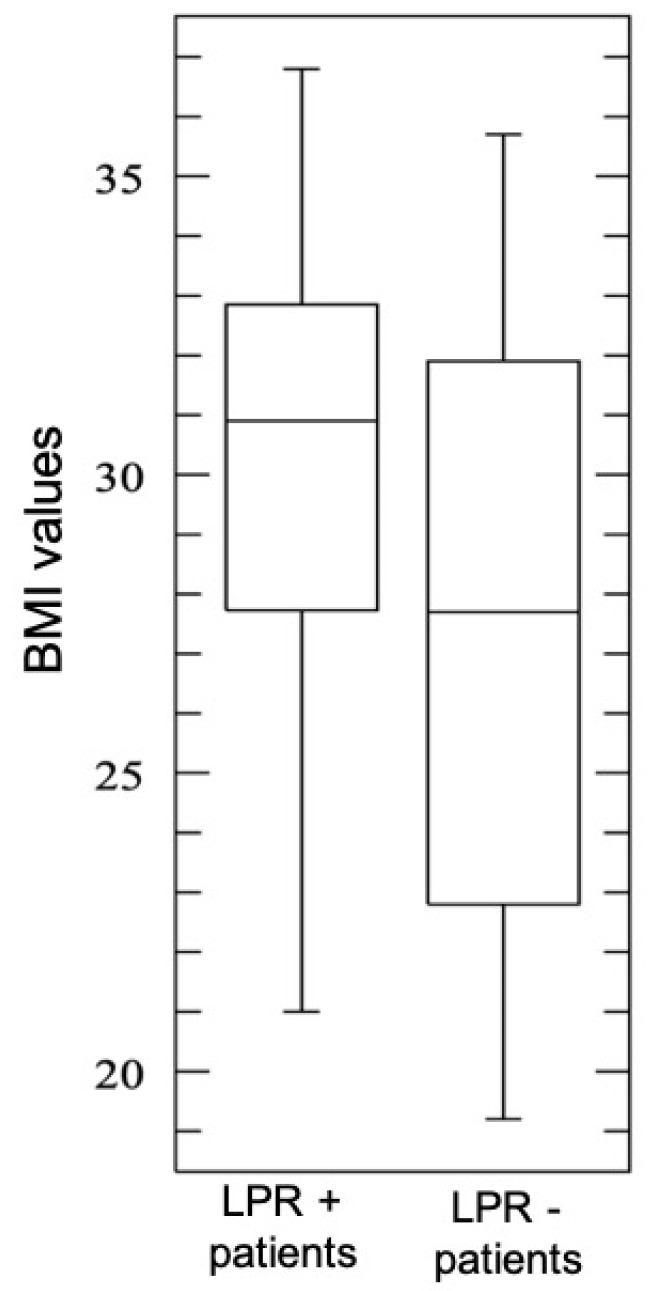
Box plot: Correlation between BMI (body max index) values of LPR (laryngopharyngeal reflux) + and LPR− patients.

**Figure 6 ijerph-16-02056-f006:**
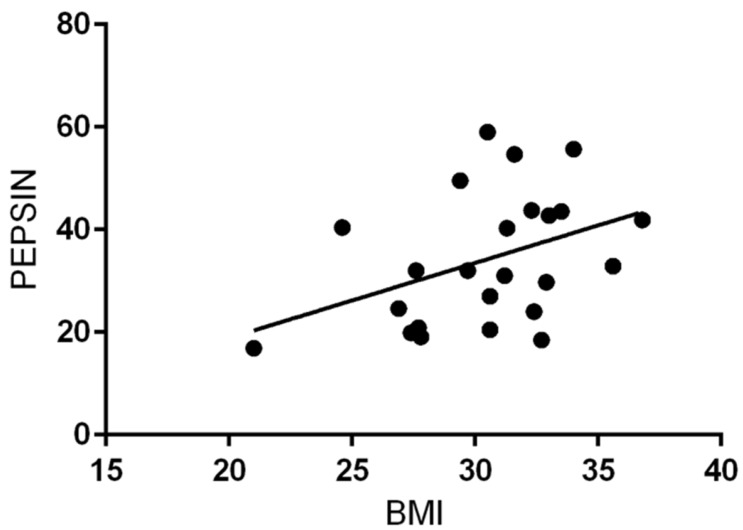
Liner regression scatterplot between BMI and salivary pepsin concentration. Statistically significant correlation (*p* = 0.05, R^2^ = 0.2).

**Table 1 ijerph-16-02056-t001:** Differences between LPR+ and LPR− patients regarding age, AHI, and BMI. Diagnosis of LPR performed using salivary pepsin dosage.

	LPR+ Patients (*n* = 24) Average Value	LPR− Patients (*n* = 51) Average Value	*p*-Value (T-Student Test)
**Age**	51.5 (range 23–75)	50.2 (range 19–66)	0.7
**AHI** (average value)	27.1 (range 10.3–54.6)	23.4 (range 8.5–47.1)	0.2
**BMI** (average value)	30.4 (range 19.9–36.8)	27.7 (range 33.8–18.5)	**0.02**

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
