# Peer review of "Laryngopharyngeal Reflux Diagnosis in Obstructive Sleep Apnea Patients Using the Pepsin Salivary Test"

_ijerph, 2019, doi:10.3390/ijerph16112056_

Round 1
Reviewer 1 Report
The investigators describe a study to determine the prevalence of laryngopharyngeal reflux among patients with OSA. Laryngopharyngeal reflux was diagnosed on the basis of a new test to determine the pepsin concentration in a salivary sample. The findings show that reflux is present in a significant proportion of patients with OSA, but that there is no correlation between the severity of OSA (as measured by AHI) and the severity of LPR (as measured by pepsin concentration). Although this study has some merit, the methods by which the study was conducted are unclear; further detail is required to make sense of how the measurements were taken and how statistics were conducted. In addition, it seems that the authors have not conducted other important statistical analyses that could be useful in understanding the association between LPR and OSA.
Major comments:
1. It is not clear from the methods whether patients were newly diagnosed with OSA or had an existing diagnosis. The authors should comment on the amount of time elapsed between AHI measurement and PEP-test measurement. If the OSA was diagnosed years ago, then the patient's current AHI may not reflect AHI at time of diagnosis, and this may explain why a strong correlation between AHI and LPR was not shown.
2. It is also not clear from the methods whether OSA patients were currently undergoing treatment (e.g., with CPAP) or were untreated. If patients were already undergoing treatment, then it may explain why no correlation was found.
3. It is unclear what is meant by "patients with pathological scores at only one of these two investigations were included from the study" [line 87-88] Isn't one aim of the study to investigate the proportion of patients with OSA who have LPR; thus, shouldn't all these subjects still be included? Is it more accurate to say that you only diagnosed patients with LPR if they had pathological scores on both the RSI and RFS?
4. It is not clear why the authors conducted linear regression between AHI and pepsin concentration (which was not significant) but not between AHI and other reflux measures, such as RSI or RFS, which may show significance.
5. The statistical analysis section needs more detail. The abstract suggests that linear regression explored the correlation between pepsin and AHI, but AHI is not mentioned in the statistical plan. It is also not clear what two groups were compared using the t-test. The statistical plan should indicate how LPR+ and LPR- patients were determined.
Minor comments:
1. I believe the authors are using the word "incidence" when they mean "prevalence".
2. I believe the authors are using the word "dosage" when they mean "concentration", e.g., salivary pepsin concentration instead of salivary pepsin dosage.
3. In general, the English language usage throughout the manuscript could use some improvement.
4. Line 91: please be more specific what is meant by "early hours of the morning" by providing a range. In addition, please indicate whether subjects were supine or upright prior to the saliva sample.
Author Response
REVIEWER 1
The investigators describe a study to determine the prevalence of laryngopharyngeal reflux among patients with OSA. Laryngopharyngeal reflux was diagnosed on the basis of a new test to determine the pepsin concentration in a salivary sample. The findings of the study show that reflux is present in a significant proportion of patients with OSA, but that there is no correlation between the severity of OSA (as measured by AHI) and the severity of LPR (as measured by pepsin concentration). Although this study has some merit, the methods by which the study was conducted are unclear; further detail is required to make sense of how the measurements were taken and how statistics were conducted. In addition, it seems that the authors have not conducted other important statistical analyses that could be useful in understanding the association between LPR and OSA.
Response to the reviewer. We have completely revised the text to clarify how the study was conducted and the criteria adopted for patient enrolment. We have specified how the measurements were made and how statistics were conducted. Other statistical analyses useful for understanding the association between LPR and OSA have been performed and results added to the text.
Major comments:
1. It is not clear from the methods whether patients were newly diagnosed with OSA or had an existing diagnosis. The authors should comment on the amount of time elapsed between AHI measurement and PEP-test measurement. If the OSA was diagnosed years ago, then the patient's current AHI may not reflect AHI at time of diagnosis, and this may explain why a strong correlation between AHI and LPR was not shown.
Response to the reviewer: all patients in the study had a recent diagnosis of OSA performed with a PSG examination. We have completely rewritten the materials and methods section in order to clarify previously unclear aspects (Page 2 lines 67-76).
2. It is also not clear from the methods whether OSA patients were currently undergoing treatment (e.g., with CPAP) or were untreated. If patients were already undergoing treatment, then it may explain why no correlation was found.
Response to the reviewer: None of the patients of the study were undergoing treatment for OSAS with a CPAP device or other medical devices. Patients who had undergone previous sleep apnea surgery were also excluded from the study. We have specified all these aspect (exclusion criterias) in the materials and methods section of the text (Page 2 lines 78-82).
3. It is unclear what is meant by "patients with pathological scores at only one of these two investigations were included from the study" [line 87-88] Isn't one aim of the study to investigate the proportion of patients with OSA who have LPR; thus, shouldn't all these subjects still be included? Is it more accurate to say that you only diagnosed patients with LPR if they had pathological scores on both the RSI and RFS?
Response to the reviewer: The original sentence reported: ‘Patients with pathological scores at only one of these two investigations were EXCLUDED from the study’[lines 87-88]. We have enrolled in the study only those patients who were positive to RSI and RSF score. Due to the many weaknesses of these clinical tests in this study we preferred to classify as clinical positive for LPRD those patients with pathological scores on both RSI and RFS. We have clarified this aspect in the text (Page 3 lines 108-112).
4. It is not clear why the authors conducted linear regression between AHI and pepsin concentration (which was not significant) but not between AHI and other reflux measures, such as RSI or RFS, which may show significance.
Response to the reviewer: We performed linear regression analysis between RSI/RFS and AHI. Linear regression didn’t show a correlation between values of AHI and RSI (p=o.4) or AHI and RSF (p=0.6). These data seem to confirm the previous results reported in literature [12,13,16,18], regarding the absence of any statistical difference regarding the AHI value between LPR + patients and LPR- patients. We have clarified this aspect in the text (Page 5 lines 212-213).
5. The statistical analysis section needs more detail. The abstract suggests that linear regression explored the correlation between pepsin and AHI, but AHI is not mentioned in the statistical plan. It is also not clear what two groups were compared using the t-test. The statistical plan should indicate how LPR+ and LPR- patients were determined.
Response to the reviewer: we have completely revised the statistical analysis section. we have clarified all the suggested aspects. (Page 4 lines 160-170).
Minor comments:
1. I believe the authors are using the word "incidence" when they mean "prevalence".
Response to the reviewer: we have changed in the text the term ‘’incidence’’ with the most appropriate ‘’prevalence’’ as suggested.
2. I believe the authors are using the word "dosage" when they mean "concentration", e.g., salivary pepsin concentration instead of salivary pepsin dosage.
Response to the reviewer: we have changed it into the test as suggested
3. In general, the English language usage throughout the manuscript could use some improvement.
Response to the reviewer: we have revised the English language as suggested.
4. Line 91: please be more specific what is meant by "early hours of the morning" by providing a range. In addition, please indicate whether subjects were supine or upright prior to the saliva sample.
Response to the reviewer: Saliva sample of patients enrolled in the study was taken between 7:00 – 8:00 a.m. in upright position. We have modified the text in order to specify the materials and methods employed (Page 3 line 125).

Reviewer 2 Report
The authors investigate the presence of laryngopharyngeal reflux in patients with OSA using the salivary pepsin dosage. The topic seems interesting, but the paper needs to be rewritten and a lot of points need to be clarified.
I would advise authors to look into the language, both in form and spelling. There are a lot of mistakes that have to be corrected. Please explain the acronyms and write all the information in the table’s captions.
Abstract
-capital letter after :
-spaces missing after .
-use of acronym without explanation (i.e. LPR)
Introduction
-clarify AHI
-AS, s not capital
-line 55 space missing
Methods
-line 67 index twice
-PSG explain
-LPRD explain
-insert definition of OSA and AHI
-line 77 the
-line 79 ‘The score was considered pathological when it was ≥13.’ Is it a universal classification?
-line 87 double space
-line 95 please add some specification of the kit, where did you buy it? What is its sensibility and accuracy?
-line 98 is the centrifugation of the sample made at room temperature?
-line 110 space missing
-explain better the kit for the pepsin dosage (is it an immunohistochemical test?) and add a picture of a sample tested with result
-line 108 Pep-test Cube please explain this test
-lines 113-116 please move these lines at the beginning of the methods
-line 119 pepsin concentration
Results
-line 126 I think that you have to use the median instead of mean. The point is how are your measures distributed? Would the median be better? If not, why?
-line 130 tab1
-line 143 space missing
-please clarify your results
-tables are not well written and clear
-overlapping of information between tables and text: you don’t need to repeat information, just present it once (either in the table or in the text)
-the statistical approach is not well clarified. Please add histograms of RSI, RFS, Pepsin concentration
-the control is missing, please explain. The idea is: what could you say of this all analysis in people not suffering from OSA?
-BMI could be the only relevant aspects, so I would like to suggest to re run tests taking into account this factor (e.g. Controlling for BMI)
-I don’t agree with the interpretation of table 2. It doesn’t seem to me that the two groups, with respect to BMI, show that big a difference. BMI seems to be the only relevant factor in both groups
-the result’s exposition is confused and should be made clearer and on point
-it would be interesting to see a scatterplot of AHI and pepsin concentration, another of BMI and pepsin concentration and a linear model’s parameters with this model (BMI, AHI and pepsin concentration as dependent). Please add them
Discussion
-line 165 space missing
-lines 177, 181,182 space missing
-line 179 episodes
-PPI explain
-line 191 double space
-lines 204-205 explain GERD and PPI
-line 206 find a synonymous of limited
-I think that you could improve the discussion
Conclusion
-the conclusion is a very strong statement only partially supported by the data presented. Please re write.
Author Response
REVIEWER 2
Abstract
1. capital letter after :
2. spaces missing after .
3. use of acronym without explanation (i.e. LPR)
Response to the reviewer: we have corrected the text as suggested.
Introduction
1. clarify AHI
Response to the reviewer: AHI definition has been clarified in the text as suggested. (Page 1 lines 36-38).
2. AS, s not capital
3. line 55 space missing
Response to the reviewer: we have corrected the text as suggested.
Methods
1. line 67 index twice
Response to the reviewer: we have corrected the text as suggested.
2. PSG explain
Response to the reviewer: we have corrected the text as suggested.
3. -LPRD explain
Response to the reviewer: we have corrected the text as suggested.
4. insert definition of OSA and AHI
Response to the reviewer: AHI and OSA definitions have been added to the text. (Page 1 lines 33-34 and lines 36-38)
5. line 77 the
Response to the reviewer: we have corrected the text as suggested.
6. -line 79 ‘The score was considered pathological when it was ≥13.’ Is it a universal classification?
Response to the reviewer: a pathological scores ³ 13 has been indicated by Belfasky et al. that first proposed the RSI score. (Belafsky PC, Postma GN, Koufman JA. Validity and reliability of the Reflux. Symptom Index (RSI). J Voice. 2002; 16: 274–7)
Different recent studies regarding LPR have been considered and validate a value ³13 indicative of LPR:
· Li J1, Zhang L2, Zhang C3, et al Linguistic Adaptation, Reliability, Validation, and Responsivity of the Chinese Version of Reflux Symptom Index. J Voice. 2016 Jan;30(1):104-8.
· Lechien JR1, Huet K2, Finck C3, et al. Validity and Reliability of a French Version of Reflux Symptom Index. J Voice. 2017 Jul;31(4):512.e1-512.e7.
· Peter C. Belafsky , MD, PhD , and Catherine J. Rees , MD. Laryngopharyngeal Refl ux: The Value of Otolaryngology Examination. Current Gastroenterology Reports 2008, 10: 278 – 282
7. line 87 double space
Response to the reviewer: we have corrected the text as suggested.
8. line 95 please add some specification of the kit, where did you buy it? What is its sensibility and accuracy?
Response to the reviewer: specification of the PEP-test kit, its sensibility and accuracy has been reported in the material and method section as suggested (page 3 lines 131-136).
9. line 98 is the centrifugation of the sample made at room temperature?
Response to the reviewer: the saliva sample was centrifuged at room temperature at 400 rpm for 5 minutes. We have specified this aspect in the text. (page 3 lines 138)
10. line 110 space missing.
Response to the reviewer: we have corrected the text as suggested.
11. explain better the kit for the pepsin dosage (is it an immunohistochemical test?) and add a picture of a sample tested with result
Response to the reviewer: we have modified the text in order to better explain the kit for the pepsin dosage. We have added a picture of a sample tested with result (Fig. 1) (Page 4 lines 143-148).
12. line 108 Pep-test Cube please explain this test
Response to the reviewer: Pep test cube is a small electronic lateral flow device that displays the result of pepsin concentration analysis in different fluids directly in ng/ml in just few seconds. We have explained the test in the materials and methods section and added a picture (Fig. 2) (Page 4 lines 149-155).
13. lines 113-116 please move these lines at the beginning of the methods
Response to the reviewer: we have rewritten the materials and methods section and moved the sentence to a specific paragraph entitled ‘Subjects of the study’.
14. -line 119 pepsin concentration
Response to the reviewer: we have deleted the sentence.
Results
1. -line 126 I think that you have to use the median instead of mean. The point is how are your measures distributed? Would the median be better? If not, why?
Response to the reviewer: We have calculated the median of RSI and RFS values of the study groups. We have modified results as suggested. (page 4 lines 180-182).
2. line 130 tab1
Response to the reviewer: we have corrected the text as suggested
3. line 143 space missing
Response to the reviewer: we have corrected the text as suggested
4. -please clarify your results
Response to the reviewer: we have modified the results section of the manuscript in order to better clarify our results
5. -tables are not well written and clear
Response to the reviewer: we have deleted Table 1 and 3 in order to avoid overlapping of information between tables and text. We have re-written table 2 in order to make it more clear.
6. -overlapping of information between tables and text: you don’t need to repeat information, just present it once (either in the table or in the text)
Response to the reviewer: we have deleted Table 1 and 3 and re-written table 2 in order to avoid overlapping of information between tables and text
7. -the statistical approach is not well clarified. Please add histograms of RSI, RFS, Pepsin concentration
Response to the reviewer: we have completely revised the statistical analysis section. we have clarified all the aspects of statistical analysis. (Page 4 lines 162-170) We have added an histograms reporting RSI, RFS, Pepsin concentration (Fig.3)
8. -the control is missing, please explain. The idea is: what could you say of this all analysis in people not suffering from OSA?
Response to the reviewer: different authors have already reported data about LPR evaluation using the salivary pepsin dosage in patients without OSA (see reference 23-31). However, we concord that a control group would have made it possible to compare the LPR prevalence, diagnosed with PEP-test, in patients with and without OSA.
Unfortunately, it was not possible to carry out a control group of patients without OSAS due to the not negligible cost of the PEP test kit. Further studies are under way to evaluate the differences in the incidence of LPR between patients with and without OSA using the salivary pepsin dosage.
9. -BMI could be the only relevant aspects, so I would like to suggest to re run tests taking into account this factor (e.g. Controlling for BMI)
Response to the reviewer: we have reviewed the statistical analysis according to the BMI results (page 5 lines 218-222). We have added a plot regarding the statistical difference about BMI between the two groups (Fig. 4)
10. -I don’t agree with the interpretation of table 2. It doesn’t seem to me that the two groups, with respect to BMI, show that big a difference. BMI seems to be the only relevant factor in both groups
Response to the reviewer: we have modified the tables of the text as suggested. We have rerun the statistical analysis (t’student test) between the two groups and we confirm the previous results with a statistical difference between LPR + and LPR- negative patients regarding average BMI. We have added a plot to better show this difference ( Fig. 5). Considering the limited range of possible BMI values, even small differences in the average values of two groups would have a statistical difference.
11. -the result’s exposition is confused and should be made clearer and on point
Response to the reviewer: we have completely revised the results section trying to make it clearer. We have reported it in points.
12. -it would be interesting to see a scatterplot of AHI and pepsin concentration, another of BMI and pepsin concentration and a linear model’s parameters with this model (BMI, AHI and pepsin concentration as dependent). Please add them
Response to the reviewer: we have added:
1) A linear regression between AHI and salivary pepsin concentration (Fig. 4).
2) A linear regression between BMI and salivary pepsin concentration (Fig. 6).
1. line 165 space missing
2. lines 177, 181,182 space missing
3. line 179 episodes
Response to the reviewer: we have corrected the text as suggested
4. PPI explain
Response to the reviewer: we have corrected the text as suggested
5. line 191 double space
Response to the reviewer: we have corrected the text as suggested
6. lines 204-205 explain GERD and PPI
Response to the reviewer: we have corrected the text as suggested
7. line 206 find a synonymous of limited
Response to the reviewer: limited was changed with restricted
8. I think that you could improve the discussion
Response to the reviewer: we have revised and try to improve the discussion section
Conclusion
1. the conclusion is a very strong statement only partially supported by the data presented. Please re write.
Response to the reviewer: the conclusion has been re-written. (Page 7 lines 300-303)

Round 2
Reviewer 1 Report
The authors have made significant changes that have resolved many of my questions, however, there remain some minor concerns:
The concluding sentence of the abstract has been revised, but is incomplete.
I appreciate that the authors have included all the individual data for the LPR-positive patients. The quality of the figure is low, however, and needs improvement. The authors also should include a legend to indicate what the different colored bars correspond to. There appear to be some subjects who have no gray-colored bar; does this mean that the pep test was not given, or that the result was 0?
For all the linear regression results, the R-squared value should be reported in addition to the p-values; although some of these fits are statistically significant based on p-value, I expect the R-squared values are quite low.
In figure 3, the box plots would be easier to interpret if the boxes were labeled with the name of each group rather than A and B. A label on the y-axis is also needed to indicate that the values represent BMI.
English language usage still needs significant improvement, both in the newly added text and the original text. There are many typos, spelling errors, and grammatical issues, which reduces the overall readability of the manuscript.
Author Response
- The concluding sentence of the abstract has been revised, but is incomplete.
RESPONSE TO THE REVIEWER: we have revised the conclusion sentence of the abstract section as suggested.
- I appreciate that the authors have included all the individual data for the LPR-positive patients. The quality of the figure is low, however, and needs improvement. The authors also should include a legend to indicate what the different colored bars correspond to. There appear to be some subjects who have no gray-colored bar; does this mean that the pep test was not given, or that the result was 0?
RESPONSE TO TE REVIEWER: we have improved the quality of the image relative to the histogram. The image has a good resolution and its final quality depends on the final size during publication. We have added the legend of the figure as suggested.
Patients with only RSI and RFS columns were those who showed positivity to clinical investigation but negativity for salivary pepsin presence (negative PEP-test). We have clarified this aspect in the legend section
- For all the linear regression results, the R-squared value should be reported in addition to the p-values; although some of these fits are statistically significant based on p-value, I expect the R-squared values are quite low.
RESPONSE TO TE REVIEWER: we have added the values corresponding to R2 for all the regression analyses carried out as suggested.
- In figure 3, the box plots would be easier to interpret if the boxes were labeled with the name of each group rather than A and B. A label on the y-axis is also needed to indicate that the values represent BMI.
RESPONSE TO TE REVIEWER: we have modified figure 3 as suggested.
- English language usage still needs significant improvement, both in the newly added text and the original text. There are many typos, spelling errors, and grammatical issues, which reduces the overall readability of the manuscript.
RESPONSE TO TE REVIEWER: we have improved English language as suggested
